# Hoof Impact and Foot-Off Accelerations in Galloping Thoroughbred Racehorses Trialling Eight Shoe–Surface Combinations

**DOI:** 10.3390/ani12172161

**Published:** 2022-08-23

**Authors:** Kate Horan, James Coburn, Kieran Kourdache, Peter Day, Henry Carnall, Liam Brinkley, Dan Harborne, Lucy Hammond, Mick Peterson, Sean Millard, Thilo Pfau

**Affiliations:** 1Department of Clinical Science and Services, The Royal Veterinary College, Hawkshead Lane, Brookmans Park, Hertfordshire AL9 7TA, UK; 2James Coburn AWCF Farriers Ltd., 8 Moulton Road, Newmarket CB8 8DU, UK; 3The British Racing School, Snailwell Road, Newmarket CB8 7NU, UK; 4Department of Biosystems and Agricultural Engineering, University of Kentucky, Lexington, KY 40506-0503, USA; 5Faculties of Kinesiology and Veterinary Medicine, University of Calgary, 2500 University Dr NW, Calgary, AB T2N 1N4, Canada

**Keywords:** racehorse, hoof, acceleration, gallop, shoeing, surface, stride time

## Abstract

**Simple Summary:**

To achieve optimal performance and low injury occurrence in horse racing, it is important to understand hoof–surface interactions. This study measured hoof accelerations in retired Thoroughbred racehorses as they galloped over turf and artificial surfaces in four shoeing conditions (aluminium, barefoot, steel and GluShu), using hoof-mounted accelerometers. During hoof landing, accelerations were increased for hindlimbs and leading limbs and on turf compared to the artificial surface. Barefoot hooves experienced the lowest impact accelerations and contrasted most with steel. During the propulsive stage of the stride, accelerations at “foot-off” were increased for low stride times, particularly in the hindlimbs, and on the artificial track. Increased impact accelerations on turf and in shod conditions could be detrimental to health and have implications for musculoskeletal injuries, whereas increased foot-off accelerations on the artificial surface may reflect this surface returning energy to the hoof and aiding propulsion, which could confer a performance benefit. Further work is needed to relate these findings to injury risk and racing outcomes specifically, particularly in racehorses galloping at top speeds.

**Abstract:**

The athletic performance and safety of racehorses is influenced by hoof–surface interactions. This intervention study assessed the effect of eight horseshoe–surface combinations on hoof acceleration patterns at impact and foot-off in 13 galloping Thoroughbred racehorses retired from racing. Aluminium, barefoot, GluShu (aluminium–rubber composite) and steel shoeing conditions were trialled on turf and artificial (Martin Collins Activ-Track) surfaces. Shod conditions were applied across all four hooves. Tri-axial accelerometers (SlamStickX, range ±500 g, sampling rate 5000 Hz) were attached to the dorsal hoof wall (x: medio-lateral, medial = positive; y: along dorsal hoof wall, proximal = positive; and z: perpendicular to hoof wall, dorsal = positive). Linear mixed models assessed whether surface, shoeing condition or stride time influenced maximum (most positive) or minimum (most negative) accelerations in x, y and z directions, using ≥40,691 strides (significance at *p* < 0.05). Day and horse–rider pair were included as random factors, and stride time was included as a covariate. Collective mean accelerations across x, y and z axes were 22–98 g at impact and 17–89 g at foot-off. The mean stride time was 0.48 ± 0.07 s (mean ±2 SD). Impact accelerations were larger on turf in all directions for forelimbs and hindlimbs (*p* ≤ 0.015), with the exception of the forelimb z-minimum, and in absolute terms, maximum values were typically double the minimum values. The surface type affected all foot-off accelerations (*p* ≤ 0.022), with the exception of the hindlimb x-maximum; for example, there was an average increase of 17% in z-maximum across limbs on the artificial track. The shoeing condition influenced all impact and foot-off accelerations in the forelimb and hindlimb datasets (*p* ≤ 0.024), with the exception of the hindlimb impact y-maximum. Barefoot hooves generally experienced the lowest accelerations. The stride time affected all impact and foot-off accelerations (*p* < 0.001). Identifying factors influencing hoof vibrations upon landing and hoof motion during propulsion bears implication for injury risk and racing outcomes.

## 1. Introduction

Whole horse kinematics and injury mechanics are influenced by hoof–surface interactions. Establishing factors that control the timing and patterns of equine hoof motion throughout a stride cycle is necessary for optimising equine biomechanical function and performance, lessening the risk of injuries and enhancing economic gains. Thoroughbred racehorses galloping at high speeds during training and racing are particularly vulnerable to injuries [1,2,3,4,5]. As a result, the racing industry are placing increasing emphasis on understanding intrinsic and extrinsic factors that modulate a horse’s output on the track. This study focuses on the latter, and, in particular, seeks to better understand how the hoof kinematics of Thoroughbred racehorses relates to their shoeing condition and the ground surface they are travelling over at a range of gallop speeds.

Horses move asymmetrically over the course of a gallop stride cycle, so the sequence of footfalls influences the accelerations and loads experienced [6,7]. Hooves experience high deceleration and impact shock vibrations as they collide with the ground surface during galloping [8]. Impact-related shock is damped by the musculoskeletal structures of the limbs and hooves [9,10,11], and the associated vibrations may be quantified by accelerometers mounted to the hooves and/or the distal limbs [12,13,14,15,16,17,18]. The accelerations just before impact can be attributed to the variability in hoof position in preparation for contact with the ground surface [16]. Larger magnitude peaks are usually caused by the impact and hoof sliding and decelerating on the surface [19]. The sliding and/or sinking into the surface lowers the forces during deceleration [20,21] and reduces bending moments on the cannon bone [19]. The primary impact, or landing phase, for each hoof is followed by the secondary impact, during which time the hoof becomes largely fixed to the ground surface. The mass of the horse and jockey moves forward in the secondary impact, and although forces experienced by the limb in question are high, the hoof experiences minimum deceleration [8,13,22]. Each limb experiences peak vertical load as the horse transitions from braking to propulsion and its centre of mass is accelerated forward [23]. This stage is termed “mid-stance” and is a period of greater hoof stability [24]. After mid-stance, during propulsion, the heels of a hoof lift away from the surface and rotate through an angle of approximately 90 degrees about the toe, and the associated limb is gradually unloaded [8,25,26]. Finally, a limb will enter the swing phase. Hoof accelerations are, once again, high due to rapid hoof rotation as the joints of the digit flex and the limbs catch up and overtake the position of the upper body.

Micro-fractures in subchondral bone, cartilage breakdown and joint degeneration have been documented in response to hoof landing during locomotion [19,27,28,29]. Some injuries have been linked to the magnitude of impact forces and surface hardness [30,31]. This includes damage to the superficial digital flexor tendons in trotters through altered loading and joint kinematics [32,33] and osseous changes associated with high-frequency vibrations, which may ultimately result in lameness [34,35]. The influence of surface type on racing injuries, specifically, has been documented previously, but with conflicting results. A pooled analysis of multiple studies suggested that there were no differences in catastrophic musculoskeletal injuries between turf and all-weather or synthetic tracks, or turf and dirt tracks [1]. However, the nuances of individual racing settings may be an important consideration. For example, the competitive nature of the track, field size, distance and prize money all vary across racing settings [36]. It may also be important to consider front and hind limbs and lead versus non-lead limbs separately. For example, when turf, synthetic and dirt tracks are compared, forelimb injuries are more common on dirt, whereas fatal hindlimb fractures are most likely to occur on turf; however, regardless of the surface, forelimbs are more likely to fracture [37]. Furthermore, surface conditions and composition must be considered; for example, temperature and moisture content influence properties such as firmness and cushioning [38,39,40]. Some research suggests that turf tracks that are faster increase the risk of fatal and non-fatal fractures and musculoskeletal injuries [2,41,42,43,44,45,46,47,48,49]. The type of injury is also a relevant consideration. For example, generally speaking, fatal and non-fatal fractures may be at a lower risk on synthetic tracks compared to turf and dirt [50]. However, the risk of biaxial proximal sesamoid bone fracture [51], fatal distal limb fracture [47,48] and lateral condylar fracture [52] may be increased on all-weather surfaces compared to turf. 

Epidemiological evidence suggests that, in addition to surface conditions, certain types of horseshoes are associated with a higher risk of racehorse injury, and hence are a further key component of the hoof–surface interaction. For example, rim shoes that are similar to natural hoof shape may decrease injury risk [53], but some studies have associated toe grabs with injury occurrence [53,54,55]. In addition, the typically flat-foot, low-heel hoof conformation of racing Thoroughbreds [56,57] has been linked to injury [58,59,60], perhaps because these horses experience different foot mechanics to other horses [61,62]. However, limited data are available to understand the effect of different shoes on racehorse gait kinematics, and studies have, so far, tended to rely upon simulated conditions, such as treadmill work [63] or mechanical shoe-testing devices [64], with unclear representation to a horse racing on a track. This is in contrast to other equestrian disciplines where the influence of shoe shape, mass, composition and other modifications have been explored more readily [65,66,67,68]. This is likely a reflection on the tightly controlled shoeing conditions permitted by the racing industry [69]. However, our recent work assessing galloping racehorses in the field has alluded to the influence shoe–surface conditions have on both hoof breakover patterns [26] and upper body movements of horses and their jockeys [70]. Further work is needed to establish the impact of shoes and surfaces on racehorse gait kinematics and kinetics, including the accelerations and loads experienced. These findings are likely to have implications for the likelihood of horse and jockey injury and falls, as well as performance. 

The aim of this study was to investigate the hoof accelerations of galloping Thoroughbreds trialling four different shoeing conditions at gallop on turf and artificial surfaces. We hypothesised that the magnitude of impact accelerations would increase on turf compared to the artificial surface, due to greater surface hardness. We also predicted that the harder the shoe composition, the higher the impact vibrations would be, and we expected shod conditions to be associated with higher impact accelerations compared to barefoot. We hypothesised that hoof accelerations would be higher at foot-off on the artificial track because breakover has previously been shown to be faster on this surface [26]. We also expected foot-off accelerations would be higher when the horses were barefoot, because unshod hooves should deform more readily upon impact and return greater energy to the hoof during propulsion.

## 2. Materials and Methods

### 2.1. Ethics

Ethical approval for this study was received from the RVC Clinical Research Ethical Review Board (URN 2018 1841-2). Informed consent was given by the jockeys, farriers and owners of the horses participating in this study. 

### 2.2. Horse and Jockey Participants

Thirteen retired Thoroughbred horses in regular work and utilised for jockey education at the British Racing School (BRS) in Newmarket, UK, provided a convenience sample. All horses were considered sound by the jockey, farriers and BRS management prior to data collection. They ranged in age from 6 to 20 years old, had heights between 15.3 and 16.3 hh (1.6–1.7 m) and their masses, quantified using a weigh tape, ranged from 421 to 555 kg. Additional body dimensions, hoof morphometrics and shoe masses for the horses are reported in Reference [71]. Four jockeys participated in this study. One horse was ridden by two jockeys, giving rise to 14 horse–jockey pairs. The same horse-and-jockey pairings were used throughout this study, so the “horse–jockey combination” was fixed, while the shoe–surface condition varied. Unfortunately, not all horse–jockey pairings completed trials in all conditions for the following reasons: (1) turf access restrictions were occasionally imposed by the BRS when this surface was considered to be very hard; (2) two of the jockeys initially recruited for the study left the racing school or were injured and unavailable to complete data collection with the horse they had been paired with; and (3) one horse died, and another was placed on rest during the period of data collection, both for reasons unrelated to this study.

### 2.3. Trial Conditions

Horses underwent trials on an artificial (Martin Collins Activ-Track, Martin Collins Enterprises Berkshire, UK) and turf surface at the BRS in four shoeing conditions: aluminium raceplates (Kerckhaert Aluminium Kings Super Sound horseshoes, Stromsholm Limited, Milton Keynes, UK), barefoot, GluShu (aluminium–rubber composite shoes, Stromsholm Limited, Milton Keynes, UK) and steel shoes (Kerckhaert Steel Kings horseshoes, Stromsholm Limited, Milton Keynes, UK). The artificial track was a mixture of well-sorted quartz sand and CLOPFF fibre, and it was wax-coated. The turf track was well-drained owing to the predominantly chalk lithology beneath. The surfaces and shoeing conditions tested were selected on the basis that they would develop understanding of the currently widely used training and racing options in the UK. In addition, we sought to investigate easily accessible yet novel shoeing options, which could be adopted by racehorse farriers, trainers and owners in the future. Horses’ hooves were trimmed prior to data collection and/or the application of shoes by the farriery team (JC, HC, LB or DH). All farriers followed the same trimming procedure set out by the lead farrier (JC). The order of trials for the eight possible shoe–surface combinations was randomized. The horses underwent a warm-up period in walk, trot, canter and gallop prior to data collection. Each data trial consisted of a minimum of two runs, to generate data for the horses galloping on both leads. However, some horses participated in additional runs if they struggled to achieve the desired lead, behaved unusually in a trial (such as bucking) or if equipment fell-off and needed to be reaffixed. Full weather data on and preceding data-collection days are available in Reference [71]. In summary, in the 72 h preceding and inclusive of data collection, mean temperature was 9.8 ± 2.3 °C (±2 SE), mean rainfall was 0.2 ± 0.1 mm (±2 SE) and mean humidity was 81.5 ± 2.5% (±2 SE).

### 2.4. Equipment 

Tri-axial accelerometers (SlamStick X, Mide Technology, United States), recording at a sample rate of 5000 Hz and with a measurement range of ±500 g, were mounted to the dorsal hoof wall of each hoof in custom-made aluminium brackets (Figure 1). The brackets were glued to the hoof by using Superfast hoof adhesive (Vettec Royal Kerckhaert, The Netherlands). The accelerometers had in-built data loggers that were capable of recording continuously for up to 30 h, and their mass was 70 g. The x-axis of the accelerometers had a medio-lateral orientation (medial = positive), the y-axis was aligned along the hoof wall (proximal = positive) and the z-axis was in the dorso-palmar orientation (dorsal hoof wall = positive). Figure 1 illustrates the bracket design and accelerometer mounted to the hoof.

### 2.5. Data Processing

Approximate timings of individual gallop runs were noted during data collection. These were used to identify relevant blocks of accelerometry data, and a custom-written MATLAB script was used to extract these data. A second custom-written MATLAB script associated with a GUI was then used to identify features of interest at hoof impact and foot-off. In summary, strides were selected manually from blocks of accelerometry data and then visually enlarged. The approximate positions of impact and foot-off (as indicated in Figure 2) for the strides were then clicked on manually. These positions were used in combination with a specified search window to identify the precise “minimum” (most negative) and “maximum” (most positive) values for the three acceleration axes at impact and foot-off. A second impact peak was used to define stride time. The size of the search window was adjusted manually to cover the same features of interest, regardless of stride duration, shoeing condition or surface. The impact was taken to encompass accelerations immediately before heel strike through to early stance. It could be readily identified from the data as being represented by acceleration spikes preceding flat traces. The flat traces were representative of stance and were followed by the foot-off acceleration spikes. Low accelerations were also associated with the swing phase, but these took place over a longer time period than stance, and the hoof was generally less stable. Data in the medio-lateral axis were inverted for the right fore and right hind to achieve a configuration with the medial direction as positive. The areas under the acceleration traces at set points forward and backward from the timing of the minimum and maximum were also quantified; these were defined both in time (±5 ms, ±10 ms, ±15 ms and ±20 ms) and by percentage through the stride (±1%, ±2%, ±3%, ±4%, ±5%). The areas reflected the change in velocity of the hoof in the period immediately prior to and post impact and foot-off. The data were grouped by horse, jockey, limb, shoeing condition, surface, gallop lead and gallop run. All data were processed by the same person (KH).

A total of 41,183 strides were included in the analysis. Table 1 summarises the number available per shoe–surface combination for each horse–jockey pair.

### 2.6. Statistics

Linear mixed models were implemented in SPSS to test for significant differences in tri-axial acceleration peaks and areas under peaks at both impact and foot-off, under the different shoe and surface conditions. Shoe, surface, stride time, “shoe*surface interaction”, “shoe*stride time interaction” and “surface*stride time interaction” were defined as fixed factors, and horse–rider pair and day were defined as random factors. Stride time was also included as a covariate. Histograms of models’ residuals were plotted, and normality was confirmed. The significance threshold in all statistical tests was set at *p* < 0.05. Models included data from all strides and a subset of the data for stride frequencies ≥2 Hz; 2 Hz is approximately equivalent to 9 ms^−1^ [72], which is a speed that is consistent with slow galloping speeds [73], and should exceed canter speeds of Thoroughbreds [63,74,75].

## 3. Results

### 3.1. Overview

We first present peak acceleration data for limbs independent of shoe–surface condition to assess the general patterns in the acceleration magnitudes per axis at impact and foot-off, independent of shoe or surface type (Table 2). Figure 2 illustrates example extracts of gallop strides per shoe–surface combination. The full raw dataset is available in the Appendix A. It was apparent from the statistical models and a visual analysis of the minimum and maximum data using boxplots that differences between the entire dataset and the subset of data for stride frequencies ≥2 Hz were slight. Therefore, we focus on presenting and discussing the data from the entire dataset in the main manuscript but make all raw data and model significance results available in the Appendix A. 

To tackle the high volume of area data, a visual inspection using boxplots was helpful in identifying consistent trends amongst parameters. It was apparent that areas calculated for time windows extending further from the peaks had trends amongst shoe–surface conditions that were similar to those closer to the peaks. Therefore, we decided to focus on a short time window (5 ms), as this best represents the nature of the main peak (sharp or broad) immediately following impact or foot-off. In addition, although the data representing areas under minimum and maximum peaks were similar, it was decided that the areas under maximum peaks depicted clearer trends and were likely to be more informative because they are associated with the accelerations transferred into the hoof in the distal to proximal direction. We also considered area data after the main peaks to be of more relevance than those that preceded the main peaks. As such, we summarise results for area data at impact and foot-off in the 5 ms time window after the maximum in Appendix B (Figure A5) and include the linear mixed model results, along with the full dataset, in the Appendix A. 

### 3.2. Forelimbs

#### 3.2.1. Impact

All significance values that were output from the linear mixed models are available in Appendix A. In summary, minimum and maximum impact accelerations for forelimbs were significantly affected by the shoe; surface; stride time; and interactions between shoe and surface, shoe and stride time and surface and stride time (*p* ≤ 0.003). The exceptions were y-maximum, which was unaffected by a shoe*surface interaction (*p* = 0.125); and z-minimum, which was unaffected by the surface type (*p* = 0.106). Minimum and maximum accelerations for the leading forelimb were significantly affected by the shoe; surface; stride time; and interactions between shoe and surface, shoe and stride time and surface and stride time (*p* ≤ 0.027). The exceptions were x-maximum, which was unaffected by a surface*stride time interaction (*p* = 0.122); and z-minimum, which was unaffected by either surface (*p* = 0.083) or a surface*stride time interaction (*p* = 0.704). Minimum and maximum accelerations for the non-leading forelimb were significantly affected by the shoe; surface; stride time; and interactions between shoe and surface, shoe and stride time and surface and stride time (*p* ≤ 0.035). The exceptions were x-maximum, which was unaffected by a shoe*stride time interaction (*p* = 0.052) or a surface*stride time interaction (*p* = 0.831); x-minimum, which was unaffected by the surface*stride time (*p* = 0.799); y-minimum, which was unaffected by the shoe (*p* = 0.114) and shoe*surface (*p* = 0.094); and z-minimum, which was unaffected by the surface (*p* = 0.530). Further details and comparisons amongst the shoe and surface conditions at impact are provided in Section 3.4, Section 3.5, Section 3.6

#### 3.2.2. Foot-Off

The minimum and maximum foot-off accelerations for forelimbs were significantly affected by the shoe; surface; stride time; and interactions between shoe and surface, shoe and stride time and surface and stride time (*p* ≤ 0.024). The exceptions were x-maximum, which was unaffected by shoe*stride time (*p* = 0.082); and y-minimum, which was unaffected by surface*stride time (*p* = 0.102). The minimum and maximum accelerations for the leading forelimb were significantly affected by the shoe; surface; stride time; and interactions between shoe and surface, shoe and stride time and surface and stride time (*p* ≤ 0.037). The exceptions were x-maximum, which was unaffected by the surface (*p* = 0.602) or a surface*stride time interaction (*p* = 0.424); y-minimum, which was unaffected by the surface (*p* = 0.170); and z-maximum, which was unaffected by either shoe (*p* = 0.072) or a shoe*stride time interaction (*p* = 0.243). Minimum and maximum accelerations for the non-leading forelimb were significantly affected by the shoe; surface; stride time; and interactions between shoe and surface, shoe and stride time and surface and stride time (*p* ≤ 0.017). This was true for all axis directions. Further details and comparisons amongst the shoe and surface conditions at foot-off are provided in Section 3.4, Section 3.5, Section 3.6.

### 3.3. Hindlimbs

#### 3.3.1. Impact

Minimum and maximum impact accelerations for hindlimbs were significantly affected by the shoe, surface, stride time, and interactions between shoe and surface, shoe and stride time and surface and stride time (*p* ≤ 0.048). The exceptions were x-maximum, which was unaffected by a shoe*stride time interaction (*p* = 0.053); y-maximum, which was unaffected by the shoe (*p* = 0.053) or a shoe*stride interaction (*p* = 0.184). Minimum and maximum accelerations for the leading hindlimb were significantly affected by the shoe; surface; stride time; and interactions between shoe and surface, shoe and stride time and surface and stride time (*p* ≤ 0.029). The exceptions were y-maximum, which was unaffected by a surface*stride time interaction (*p* = 0.755); y-minimum, which was unaffected by the surface (*p* = 0.916); z-maximum, which was unaffected by a surface*stride time interaction (*p* = 0.962); and z-minimum, which was unaffected by either shoe (*p* = 0.145) or a shoe*stride time interaction (*p* = 0.067). Minimum and maximum accelerations for the non-leading hindlimb were significantly affected by the shoe; surface; stride time, and interactions between shoe and surface, shoe and stride time and surface and stride time (*p* ≤ 0.028). The exception was x-minimum, which was unaffected by a shoe (*p* = 0.267) or a shoe*stride time interaction (*p* = 0.431). Further details and comparisons amongst the shoe and surface conditions at impact are provided in Section 3.4, Section 3.5, Section 3.6.

#### 3.3.2. Foot-Off

Minimum and maximum foot-off accelerations for hindlimbs were significantly affected by the shoe; surface; stride time; and interactions between shoe and surface, shoe and stride time and surface and stride time (*p* < 0.001). The exception was x-maximum, which was unaffected by the surface (*p* = 0.144) or a surface*stride time interaction (*p* = 0.288). Minimum and maximum accelerations for the leading hindlimb were significantly affected by the shoe; surface; stride time; and interactions between shoe and surface, shoe and stride time, and surface and stride time (*p* ≤ 0.044). The exception was x-maximum, which was unaffected by a shoe*stride time interaction (*p* = 0.090). Minimum and maximum accelerations for the non-leading hindlimb were significantly affected by the shoe; surface; stride time; and interactions between shoe and surface, shoe and stride time and surface and stride time (all *p* ≤ 0.047). Further details and comparisons amongst the shoe and surface conditions at foot-off are provided in Section 3.4, Section 3.5, Section 3.6.

### 3.4. Summary of Shoeing Condition Effect

#### 3.4.1. Impact

The estimated marginal means (EMMs) for shoeing condition effects on impact are presented in Appendix A and all post hoc pairwise comparisons (with Bonferroni correction) are provided in Appendix A. In each case, where EMM differences are reported below for pairwise comparisons, the first condition mentioned has the larger EMM value resulting in a positive difference.

The shoeing condition significantly influenced all impact accelerations in the forelimbs (*p* ≤ 0.026), with the exception of y-minimum in the non-leading forelimb (*p* = 0.114). In the hindlimbs, the shoeing condition significantly influenced all impact accelerations (*p* ≤ 0.010), with the exception of y-maximum for the combined dataset (*p* = 0.053); z-minimum in the leading hindlimb (*p* = 0.145) and x-minimum in the non-leading hindlimb (*p* = 0.267). The EMM impact accelerations were largest in terms of absolute magnitude for y-maximum in the leading hindlimb (mean of EMMs across shoeing conditions was 128.1 g), with the individual largest EMM acceleration of 142.7 ± 21.9 g (mean ±2 SE in this section) being recorded for the y-maximum in the steel shoeing condition. For the y-maximum parameter, steel was most different to the barefoot condition particularly in the leading hindlimb (EMM difference = Δ33.1 ± 4.1 g) (Appendix A). Mean impact maximum accelerations were 1.8–3.1 times larger than mean impact minimum accelerations, per axis direction. Y-maximum and z-maximum accelerations were of comparable magnitude and approximately double the magnitude of the x-maximum. The smallest accelerations, in terms of absolute magnitude, were recorded for the x-minimum in the non-leading forelimb (EMM average across shoeing conditions was 19.2 g), with the individual smallest absolute EMM acceleration of 16.2 ± 5.0 g being recorded for the x-minimum in the barefoot condition. 

The largest impact acceleration offsets amongst shoeing conditions, in absolute terms, were most commonly observed between the steel and barefoot conditions (Appendix A): 11/18 comparisons in the forelimbs (combined forelimb data, leading forelimb and non-leading forelimb) and 6/18 comparisons in the hindlimbs (combined hindlimb data, leading hindlimb and non-leading hindlimb); for the six acceleration directions. Barefoot was amongst the pairwise comparisons with the largest offsets in 29/36 instances. Considering all six acceleration axes together in the individual limb datasets, the barefoot condition generated the lowest absolute means for EMM accelerations for all limbs: these ranged from 43.7 ± 20.8 g (*n* = 6) in the non-leading forelimb to 64.1 ± 30.8 g in the leading hindlimb. In contrast, the steel condition generated the highest absolute means for EMM accelerations for all limbs: these ranged from 53.1 ± 23.9 g in the non-leading forelimb to 74.2 ± 35.5 g in the leading hindlimb.

#### 3.4.2. Foot-Off

The shoeing condition significantly influenced all foot-off accelerations in the forelimbs (*p* ≤ 0.024), with the exception of z-maximum in the leading forelimb (*p* = 0.072). In the hindlimbs, the shoeing condition significantly influenced all foot-off accelerations (*p* ≤ 0.044). The estimated marginal means (EMMs) for shoeing condition effects on foot-off are presented in Appendix A. The mean foot-off accelerations were largest in terms of absolute magnitude for y-maximum in the combined forelimb dataset (mean of EMMs across shoeing conditions was 95.6 g), with the individual largest EMM acceleration of 98.5 ± 11.2 g being recorded for the y-maximum in the GluShu shoeing condition for the leading forelimb; the subsequent three largest acceleration magnitudes were for the y-maximum in the steel condition (97.2 ± 11.2 g, 98.4 ± 9.8 g and 97.6 ± 10.7 g for the leading, non-leading and combined forelimb data, respectively) The smallest accelerations, in terms of absolute magnitude, were recorded for the x-maximum in the non-leading forelimb (mean across shoeing conditions was 13.7 g), with the individual smallest absolute EMM acceleration of 13.6 ± 2.3 g being recorded for the x-maximum in the GluShu condition for the non-leading forelimb. Noticeably, the y-maximum accelerations were considerably larger than accelerations in the other axis directions: y-maximum accelerations were on average 5.4 times larger than x-axis accelerations, and 2.3 times larger than the y-minimum and z-axis accelerations. Nevertheless, pairwise comparisons between shoeing conditions only indicated a maximum difference of Δ13.1 ± 1.2 g, which occurred between steel versus barefoot for the y-minimum parameter. The mean absolute difference between EMMs for pairwise shoe comparisons across all limbs was Δ3.2 ± 0.3 g.

Considering all six acceleration axes together in the individual limb datasets, differences between conditions were small. Overall, the barefoot condition generated the lowest absolute means for EMM accelerations for all limbs: these ranged from 37.1 ± 23.9 and 37.1 ± 19.5 g (*n* = 6) in the non-leading forelimb and the leading hindlimb, respectively, to 37.3 ± 23.7 g in the leading forelimb and 39.1 ± 21.6 g in the leading hindlimb. In contrast, the steel condition generated the highest absolute means for EMM accelerations for all limbs: these ranged from 40.2 ± 25.2 g in the non-leading forelimb to 43.8 ± 23.2 g in the non-leading hindlimb. However, it is important to note that the results showed a dependency on stride time (Section 3.7).

### 3.5. Summary of Surface Effect

#### 3.5.1. Impact 

The surface type significantly influenced all impact accelerations in the forelimbs (*p* ≤ 0.013), with the exception of z-minimum (*p* = 0.106 for the combined forelimb data, *p* = 0.083 for the leading forelimb and *p* = 0.530 for the non-leading forelimb). In the hindlimbs, surface significantly affected all impact accelerations (*p* ≤ 0.015), with the exception of y-minimum (*p* = 0.916) in the leading hindlimb. All EMMs for surface effects on impact are presented in Appendix A and the post hoc results for pairwise artificial–turf comparisons are presented in Appendix A. Surface effects were clearly apparent across all acceleration axes (Figure 3; Figure A1). Impact accelerations were always larger on turf in all directions for forelimbs and hindlimbs. Of note, the z-maximum and y-maximum provoked the highest accelerations. The highest EMM acceleration was recorded on turf for y-maximum in the leading hindlimb (167.1 ± 21.8 g; mean ±2 SE in this section), while the lowest was observed for the x-minimum on the artificial in the non-leading forelimb (15.0 ± 5.0 g). The greatest contrast was present for the y-maximum in the leading hindlimb; EMM y-maximum acceleration was Δ78.1 ± 3.6 g higher on turf compared to the artificial surface. The largest difference in absolute terms for the forelimbs occurred for the leading forelimb (Δ60.7 ± 2.8 g) for the z-maximum parameter.

Considering all six acceleration axes together in the individual limb datasets, the absolute means of EMM accelerations on the artificial surface ranged from 36.4 ± 15.7 g (*n* = 6) in the non-leading forelimb to 49.5 ± 23.8 g in the leading hindlimb. In contrast, absolute means for EMM accelerations on turf ranged from 59.8 ± 28.6 g in the non-leading forelimb to 87.7 ± 41.5 g in the leading hindlimb.

#### 3.5.2. Foot-Off

The surface type significantly influenced all foot-off accelerations in the forelimbs (*p* ≤ 0.022), with the exception of x-maximum (*p* = 0.602) and y-minimum (*p* = 0.170) in the leading forelimb. In the hindlimbs, surface type significantly affected all foot-off accelerations (*p* ≤ 0.047), with the exception of x-maximum (*p* = 0.144) in combined hindlimb data. The EMMs for surface effects on foot-off are presented in Appendix A. Foot-off accelerations were nearly always greater on the artificial surface, with the exceptions being x-minimum and y-minimum in the non-leading forelimb; x-minimum in the leading hindlimb; and x-maximum and y-minimum in the non-leading hindlimb. As observed amongst the shoeing condition effects, the y-maximum accelerations were noticeably larger than accelerations in the other directions: here, y-maximum accelerations were, on average, 5.0 times larger than x-axis accelerations and 2.3 times larger than the y-minimum, z-minimum and z-maximum accelerations. However, it was again the z-maximum—in this case, for the leading forelimb—which displayed the greatest contrast (Δ12.6 ± 1.2 g) between turf and artificial surfaces (Appendix A). On average, for forelimbs, the z-maximum accelerations were 25% higher on turf, and this compared to an increase of 9% on turf for hindlimbs

Considering all six acceleration axes together in the individual limb datasets, the absolute means of EMM accelerations on the artificial surface ranged from 40.3 ± 25.3 g for the non-leading forelimb to 42.2 ± 25.7 g in the leading forelimb. Absolute means for EMM accelerations on turf ranged from 37.2 ± 22.6 g in the leading forelimb and 37.2 ± 22.5 g in the non-leading forelimb to 40.8 ± 21.0 g in the non-leading hindlimb. These data therefore indicate that the accelerations on turf at foot-off were reduced relative to the artificial surface, in contrast to the impact data where the reverse was true. Specifically, accelerations were up to 33% greater on the artificial surface compared to the turf for forelimbs and up to 20% greater in the hindlimbs.

**Figure 3 animals-12-02161-f003:**
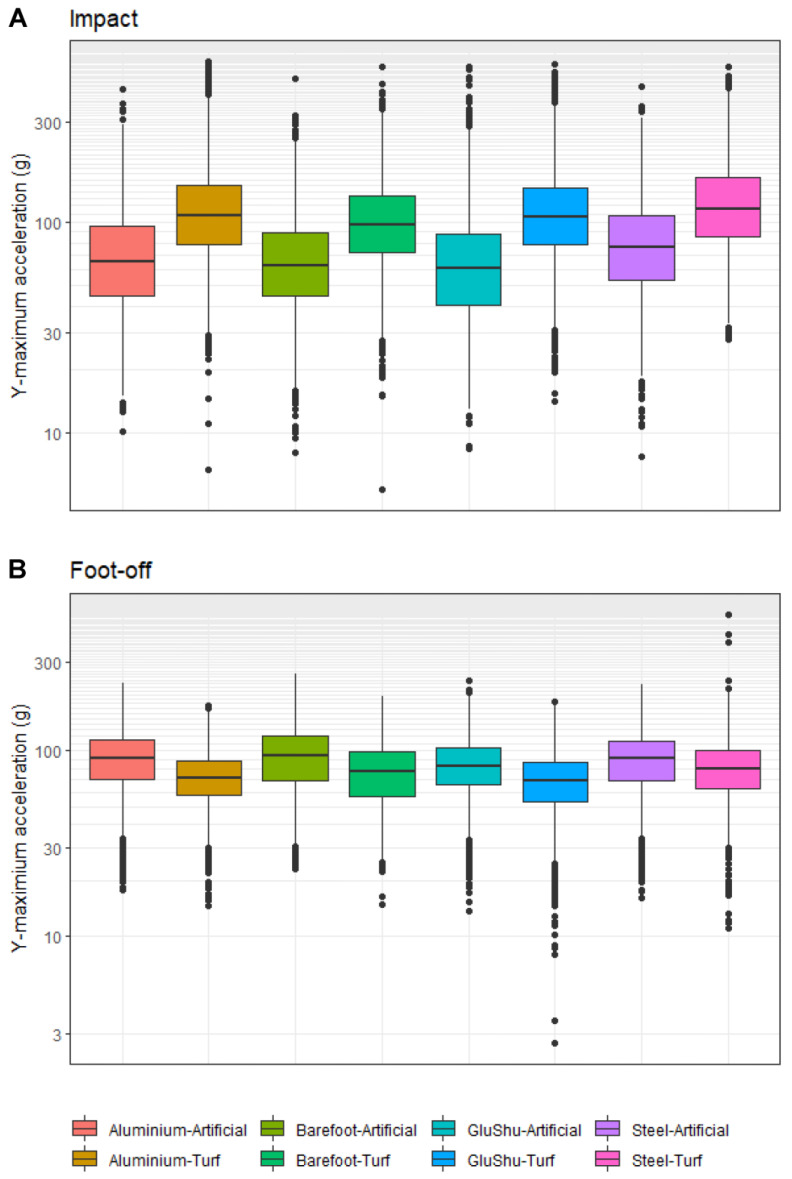
(**A**) Boxplots illustrating the impact accelerations for the y-maximum parameter across shoe–surface combinations. (**B**) Boxplots illustrating the foot-off accelerations for the y-maximum parameter across shoe–surface combinations. Data are pooled across limbs in each case.

### 3.6. Shoe–Surface Interactions

The EMMs for shoe–surface combinations are presented in Appendix A. Post hoc tests were run on acceleration parameters that indicated a significant shoe–surface interaction. As the y-maximum parameter seemed to be particularly sensitive to shoe and surface effects (Section 3.4 and Section 3.5) and often recorded the highest peak accelerations, we focus on outlining the results from this parameter. This is also the parameter most strongly correlated to stride time at foot-off (Appendix B Figure A3). However, the full output of this post hoc analysis for all acceleration parameters is provided in Appendix A. In each case, where EMM differences are reported below for pairwise comparisons, the first condition mentioned has the larger EMM value, resulting in a positive difference. Please note that values reported below, in the shoe–surface comparisons, are from the post hoc models (Appendix A) and, hence, differ slightly from those from the initial models (Appendix A).

#### 3.6.1. Impact

The y-maximum data for the leading forelimb indicated that the greatest EMM difference of Δ72.8 ± 3.8 g (mean ± 2 SE in this section) was observed between the aluminium–turf and barefoot–artificial conditions. However, this was closely followed by the comparison between the steel–turf and barefoot–artificial conditions, which had an EMM difference of Δ71.4 ± 3.9 g. The only pairwise comparisons that were not significantly different were aluminium–artificial versus GluShu–artificial, aluminium–turf versus steel–turf and barefoot–turf versus GluShu–Turf (*p* = 1.0 in each case). 

For the non-leading forelimb, the magnitudes of EMM differences were smaller. Specifically, the largest EMM difference between the steel–turf and barefoot–artificial was Δ64.3 ± 3.3 g, which was followed by the comparison between the steel–turf versus aluminium–artificial (EMM difference = Δ58.6 ± 3.5 g) and GluShu–turf versus barefoot–artificial (EMM difference = Δ58.4 ± 3.4 g). Only the aluminium–artificial and GluShu–artificial combinations were not significantly different (*p* = 0.166). Please note that the combined forelimb data were not significantly affected by a shoe–surface interaction (Appendix A), so the offsets amongst shoe–surface conditions are not reported here. 

The y-maximum data for the hindlimbs indicated that larger acceleration differences were at play amongst the shoe–surface combinations, relative to the forelimb data. For the combined hindlimb data, the EMM differences between the aluminium–turf versus barefoot–artificial and the GluShu–turf versus barefoot–artificial, each of Δ89.0 g, were the largest, followed by steel–turf versus barefoot–artificial, which had an EMM difference of Δ87.1 ± 3.8 g. There were four pairwise comparisons that were not significantly different: aluminium–artificial versus GluShu–artificial; aluminium–turf versus GluShu–turf; aluminium–turf versus steel–turf; and GluShu–turf versus steel–turf (all *p* = 1.0). All other comparisons for the combined hindlimb data were significant (*p* ≤ 0.006).

For the leading hindlimb, peak EMM acceleration differences reached Δ110.0 ± 5.7 g for the steel–turf versus barefoot–artificial condition, closely followed by GluShu–turf versus barefoot–artificial (EMM difference = Δ99.6 ± 6.0 g) and steel–turf versus aluminium–artificial (EMM difference = Δ99.4 ± 6.1 g). In contrast, the aluminium–turf versus GluShu–turf comparison indicated no significant difference (*p* = 1.0), and the GluShu–artificial and steel–artificial were also not significantly different (*p* = 0.104); all other comparisons highlighted significant differences (*p* ≤ 0.007).

The non-leading hindlimb data indicated that GluShu–turf versus barefoot–artificial was most different (EMM difference of Δ75.8 ± 5.2 g), followed by aluminium–turf versus barefoot–artificial (Δ72.8 ± 4.8 g) and steel–turf versus barefoot–artificial (Δ68.3 ± 4.8 g). The data therefore show that differences amongst shoe–surface conditions were greater in the leading limbs, for both hindlimbs and forelimbs. For the non-leading hindlimb, there were four non-significantly different conditions: aluminium–artificial versus GluShu–artificial; aluminium–turf versus GluShu–turf; aluminium–turf versus steel–turf; and GluShu–artificial versus steel–artificial (all *p* = 1.0). All other comparisons were significant (*p* ≤ 0.029). 

#### 3.6.2. Foot-Off

Accelerations at foot-off had smaller absolute magnitudes for EMM differences amongst shoe–surface combinations. For the combined forelimbs, the largest EMM difference of Δ11.1 ± 1.5 g was observed between the GluShu–artificial versus aluminium–turf, closely followed by the GluShu–artificial versus GluShu–turf (EMM difference = Δ10.7 ± 1.3 g) and the steel–artificial versus aluminium–turf (EMM difference = Δ10.2 ± 1.4 g). Only the aluminium–artificial versus barefoot–artificial, aluminium–turf versus GluShu–turf, barefoot–turf versus steel–turf and GluShu–artificial versus steel–artificial conditions were not significantly different (each with *p* = 1.00); all other pairwise comparisons were significant, with *p*-values <0.001.

For the leading forelimb, there were eight EMM differences with magnitudes exceeding Δ10 g. The largest differences were between GluShu–artificial and aluminium–turf (Δ17.3 ± 2.0 g), followed by GluShu–artificial versus barefoot–turf (Δ15.4 ± 2.2 g) and steel–artificial versus aluminium–turf (Δ14.4 ± 2.0 g). The following pairwise comparisons were not statistically different: aluminium–artificial versus barefoot–artificial (*p* = 1.00), aluminium–turf versus barefoot–turf (*p* = 1.00), barefoot–turf versus steel–turf (*p* = 0.375) and GluShu–turf versus steel–turf (*p* = 1.00); all other comparisons had *p*-values ≤0.049.

For the non-leading forelimb, the GluShu–artificial versus GluShu–turf and steel–artificial versus GluShu–turf comparisons, each had differences of Δ13.9 g. They were followed in magnitude by the barefoot–artificial versus GluShu–turf EMM difference (Δ12.4 ± 2.0 g). For this limb, there were seven pairwise shoe–surface comparisons that were not significantly different (*p* ≥ 0.084).

Amongst the combined hindlimb data, differences amongst conditions were comparable to the forelimb data. The largest EMM difference was for the barefoot–turf versus GluShu–turf (Δ12.5 ± 1.5 g), followed by the steel–artificial versus GluShu–turf (Δ11.6 ± 1.5 g). All other EMM differences were less than Δ10 g, and eight of these were non-significant (*p* = 1.00). 

EMM differences in the leading hindlimb were largest between the steel–artificial and GluShu–turf conditions at Δ16.9 ± 2.1 g. This was followed by the EMM differences between steel–artificial and aluminium–turf (Δ13.9 ± 2.1 g) and steel–artificial versus steel–turf (Δ13.8 ± 2.0 g). Here, there were more comparisons that were not significantly different; the *p*-values for 11 comparisons were ≥0.095. 

### 3.7. Stride Time

Impact accelerations for all axes showed weak negative correlations with stride time. There were moderate correlations between stride time and foot-off accelerations, in particular for the y-maximum and z-maximum parameters. Figure 4 summarises data for the y-maximum acceleration parameter, subdivided according to shoe–surface combination (using the full dataset for forelimbs and hindlimbs). These trends are further illustrated in Appendix B Figure A4, where the strength of the correlations and significance values are also indicated per limb type for all acceleration axes. 

## 4. Discussion

Identifying factors that influence the accelerations experienced by a horse’s hoof during landing and take-off bears implication for the injury risk and performance of racehorses and jockeys on the racetrack. Although epidemiological evidence indicates that multiple factors are associated with injury risk, including horse characteristics (age, sex and performance quality), training and racing history, pre-existing injuries and race characteristics (e.g., geometry and class), the ground-surface conditions and hoof-shoeing conditions are factors that may be managed relatively easily and offer practical solutions to improve racing outcomes. This study emphasised the important influence that ground-surface type and shoeing condition have on tri-axial accelerations experienced at the dorsal hoof wall. 

### 4.1. Impact 

The data presented indicated that the hoof accelerations experienced at impact were 1.4–1.9 times and 1.2–2.4 times greater on turf compared to the artificial track for the forelimbs and hindlimbs, respectively. This trend toward higher accelerations on turf than synthetic surfaces is consistent with previous studies comparing hoof impact variations on turf versus synthetic surfaces at the trot and canter [16]. The acceleration power and frequency of hoof wall vibrations have previously been linked to surface hardness [13,16,21,76,77,78]. It is likely that the results here reflect the greater hardness of the turf compared to the artificial surface, even under the turf conditions studied, which were deemed “soft” to “good to firm”. This is important to recognise, as track hardness appears to be related to racing injuries [31,79]. The artificial surface was probably better at damping impact accelerations and provided greater cushioning to the hoof upon landing, despite this not being perceivable by the jockeys [71]. This interpretation is consistent with previous studies, which have demonstrated that synthetic surfaces have a higher damping capacity and lead to reduced hoof vibrations upon landing than turf, dirt and crushed sand [16,17,80]. 

The smallest relative difference between surfaces was observed for the z-minimum in all limbs, although this parameter was the most difficult to target consistently during data processing. For the forelimbs, the y-maximum and z-maximum showed the largest relative differences between surfaces (all 1.7–1.9 times larger on turf). In the hindlimbs, the greatest relative difference between surfaces was apparent for the y-minimum (2.2–2.4 times greater on turf); however, as noted in the Results section, the largest absolute difference between surfaces was 78.1 g for the y-maximum parameter in the leading hindlimb. The differing sensitivity of the six acceleration axes to surface type between forelimbs and hindlimbs may indicate that the hoof orientation on and immediately after landing is consistently different between the forelimbs and hindlimbs. For example, it would make sense for the accelerations represented by z-maximum (in the dorsal direction of the hoof) to be most closely associated with horizontal braking. Forelimbs are responsible for decelerating the horse in each stride cycle [22,81,82], and it is therefore logical that the z-maximum parameter would have greater sensitivity to the surface type in forelimbs. In addition, the forelimb hooves may have a greater tendency to land obliquely, whereas it is possible that the hindlimb hooves adopt an orientation on landing whereby the solar surface is closer to a parallel orientation to the ground surface, which could explain the particular sensitivity of the hindlimb y-axis data to surface condition.

Previous studies have indicated that the right hindlimb on a counter-clockwise track (this would be the non-leading hindlimb) has an overall higher incidence of fracture than the left hindlimb, but it shows no difference in injury risk due to surface type [37]. In addition, previous comparisons between contralateral and ipsilateral pairs of limbs found that the leading forelimb and non-leading hindlimb were at greater risk [37]. Data from this study suggest that the leading limbs experience the largest accelerations on the tracks used here, which had only a very slight anticlockwise bend [70]. Based on the EMMs for surface effects (Appendix A) averaged across all acceleration axes, the leading limbs had accelerations that were 1.5 times larger than the non-leading limbs. This could make them more vulnerable to injury, as high accelerations reflect more rapid loading during the secondary impact phase of the stride cycle, and previous work has found a relationship between impact forces and lameness [30]. Comparing the impact data across all stride times, it is suggested from our data that, at low stride times (i.e., higher speeds), the accelerations in the dorso-palmar (especially z-maximum) direction are proportionally larger for the leading limbs (Appendix B Figure A4). This could suggest that the hoof landing patterns are also dependent on speed and may reflect the reduced overlap between individual limb-stance phases at higher gallop speeds [72]. During impact, the y-axis data (along the hoof wall) are probably most closely related to the ground reaction force, and this may explain why the accelerations were often large along this axis and why the y-maximum parameter appeared to be particularly sensitive to shoe–surface conditions. Moreover, at higher gallop speeds, it is plausible that there is proportionally greater hindlimb loading [26], and this may also help to explain why the y-minimum EMMs showed the largest proportional differences between surfaces for hindlimbs when assessing the data as a whole.

When all acceleration axes were considered together, the data indicated that impact acceleration peaks were 7–12% higher in aluminium shoes, 2–8% higher in GluShus and 10–18% higher in steel shoes when compared to the barefoot condition (comparisons made in the individual limbs). This compares to a previously reported difference of 15% between shod and unshod horses during simulated impact loading at trot in an in-vitro model [11]. However, accelerations were up to 25–30% higher in steel shoes compared to barefoot for the y-minimum and x-minimum parameters. The higher accelerations typically associated with the steel shoeing condition at impact may reflect the high rigidity and relative hardness of steel [83], which would have initiated rapid energy loss through hoof and limb vibrations. Shoeing with steel shoes has also been found to increase the maximal vertical force compared to barefoot in trotting Warmblood horses [84]. In contrast, the greater similarity in accelerations for GluShu relative to barefoot probably reflects greater damping in this condition due to the rubber coating on the shoe. Some previous work has also found synthetic polyurethane shoes and pads made of synthetic rubber can reduce peak impact vibrations in trotting horses [12,85], although no differences amongst shoeing conditions with and without a pad and packing material were found in an in vitro model [86]. The hoof was perhaps most efficient at energy absorption on landing when barefoot because the tubules embedded in the inter-tubular matrix were better able to dissipate energy through cracking and deformation while protecting the matrix from fracture or damage [87]. Unshod feet are known to undergo a greater degree of heel expansion, and this movement could help to dissipate the impact vibrations [88]. In addition, because the barefoot sole is closer to the ground surface compared to a shod foot, the frog and solear surface participate in the impact sooner than in the shod conditions, and the load will be more readily distributed over the full area of the solear surface [89]; an effect that is also likely to reduce the frequency of the vibrations measured at the dorsal hoof wall. When combined with the artificial surface, it is therefore unsurprising that the barefoot hooves usually experienced the lowest impact accelerations and typically contrasted most with steel shod hooves on turf. These findings tie in with the horses’ and jockeys’ centre-of-mass displacements. The largest vertical centre of mass displacement differences were also present between barefoot–artificial and steel–turf conditions [70], suggesting that the patterns in hoof kinematics may be translated into upper-body kinematics. 

Further work is needed to establish the relative risk of damage to the hoof and more proximal limb structures in association with the observed variability in accelerations amongst barefoot and shod conditions at gallop. It will be important to establish whether there are certain thresholds for impact accelerations that may be conducive to the development of injuries or pathologies, such as osteoarthritis. Performance implications are also key. Previous work has suggested that a reduction in the decelerative peak may signify an increased stride efficiency, by permitting a smoother transition from retardation to propulsion [78]. This may be important in determining the safety of racing surfaces also. However, although the impact accelerations were commonly lower when barefoot, it is worth noting that, when galloping barefoot on turf, a greater proportion of the runs involved the horses swapping leads (Appendix A); 18% of mixed-lead forelimb runs and 23% of mixed-lead hindlimb runs were from the barefoot–turf condition compared to just 7–9% of the data from the individual limbs. This could signify that the horses were more unbalanced in the barefoot–turf condition and may explain why the jockeys perceived gallop runs to be less smooth, most commonly variable and occasionally unsafe (17% of trials) in this condition [71]. Hoof acceleration signals at different stages of the trimming/shoeing cycle will also be important to understand. For example, a gradual dorsal shift in the centre of pressure with respect to the distal interphalangeal joint due to hoof growth and backward tilting of the foot in unshod hooves [84] may influence the depth of penetration of the heel into a compliant surface during loading. Indeed, hoof pitch rotation during early stance if horses’ heels sink into a surface has been reported at walk [90]. This effect may influence the magnitude of impact accelerations being recorded at the hoof wall. 

### 4.2. Foot-Off

At foot-off, the large acceleration spikes are caused by the hoof accelerating to the forward speed of the horse. Accelerations were more similar between turf and artificial surfaces when compared to the impact data (Figure 3, Appendix A) but were almost always larger on the artificial surface. The maximum absolute differences occurred in the leading forelimb between surfaces for the z-maximum and y-maximum (difference of 12.6 g and 10.4 g, respectively), and this may reflect the fact that, in this limb, the braking and vertical impulses must decelerate the centre of mass and provide it with sufficient upward vertical velocity for the flight phase of the stride [81]. Vertical centre of mass displacements were indeed larger by around 5.7 mm downward and 2.5 mm upward on the artificial surface as a result of this action [70] (Figure A6; Figure A7). The general trend toward higher hoof accelerations on the artificial surface is also consistent with a faster breakover on this surface [26]. Larger acceleration peaks at foot-off have previously been related to faster track rebound rates and reduced hardness [78]. Here, the more deformable artificial track may return greater energy to the hoof during the propulsive phase, leading to a more energy-efficient gait and also explaining these higher accelerations. Nevertheless, even if an artificial surface might be deemed favourable, as the majority of UK racing currently takes place on turf tracks, there may be logistical constraints in the immediate future. 

Consistent with the impact data, we found that the barefoot hoof had the lowest hoof accelerations on average across all stride times. Considering all acceleration axes together, our data indicate that foot-off accelerations were 4–7% higher in aluminium shoes, 5–6% higher in GluShus and 8–12% higher in steel shoes, when compared to the barefoot condition (comparisons made in the individual limbs). The pairwise comparisons with the largest offsets included barefoot in 18/36 cases; however, in two of these instances (for the x-minimum), the barefoot condition actually had the larger accelerations of the two shoeing conditions. In some ways, these observations are surprising because a barefoot hoof would be expected to deform more on impact and subsequently return more energy to the hoof, which might be expected to lead to more rapid accelerations. Indeed, at the higher gallop speeds, barefoot hooves do appear to experience proportionally higher foot-off accelerations relative to the shod conditions (Figure 4). This is consistent with observations that breakover becomes relatively faster for barefoot hooves at higher gallop speeds in the non-leading hindlimb; the only limb in which breakover duration was found to be sensitive to shoeing condition [26]. This previous work [26] also indicated that, at low gallop speeds, barefoot hooves have a longer breakover duration relative to the shod conditions (in the non-leading hindlimb). It was proposed that shoe shape, and in particular the bevelled toe of the shoes, might be important for increasing the breakover rate [26] and, by extrapolation here, hoof accelerations also, in the shod conditions at low–moderate speeds. As the retired ex-racehorses used in this study tended to gallop at average speeds of around 40 km h^−1^, this may explain why accelerations associated with the barefoot condition tended to be reduced on average. In addition, given it was the mixed lead data that represented more of the faster barefoot runs (Appendix A), it is consistent that an analysis of the barefoot condition in the individual limb datasets (for lower stride times) would tend toward lower values. Interestingly, it was the non-leading hindlimb that experienced slightly higher accelerations overall at foot-off compared to the other limbs (up to 6% higher on average across all acceleration axes), and this may explain its sensitivity to shoeing conditions in terms of breakover duration [26]. 

At the upper range of gallop speeds assessed here, the foot-off accelerations were increased across all acceleration axes proportionally more in the hindlimbs compared to the forelimbs, with the exception of the x-minimum in the non-leading hindlimb. This trend also mirrors the observations in the breakover duration data [26]. It likely relates to a difference in landing orientation and subsequent hoof trajectory in forelimbs versus hindlimbs as speeds increase, including greater hindlimb loading and more rapid push-off from the hind end. Further work is needed to establish the effect of turns on hoof acceleration patterns, as turning imposes additional asymmetrical forces on the limbs on the inside and outside of the turn [91]. The effect of hoof growth on foot-off accelerations is also potentially important. Long toes may lead to longer breakover durations, and due to an increase in the length of the resistance arm, there will be an increase in tension on the deep digital flexor tendon to initiate breakover [92], and toe-penetration depth is likely to increase. Extending the duration of breakover could reduce the magnitude of accelerations in the foot-off window.

## 5. Conclusions

Tri-axial hoof accelerations at impact and foot-off in galloping Thoroughbreds were influenced by the horses’ shoeing condition and surface type. Accelerations were elevated at impact on the turf surface compared to the artificial track by 1.2–2.4 times across limbs, depending on the acceleration axis considered; acceleration magnitudes were largest and offsets between surfaces greatest along the hoof wall and in the dorso-palmar direction. Accelerations were, on average, 2–18% higher at impact in the shod conditions compared to barefoot, when considering all acceleration axis directions together, but they rose up to 30% more in steel. Preventing excessive shock loading and related musculoskeletal injuries in racehorses is of critical relevance to the racing industry. This work suggests that the combination of an artificial surface and barefoot hooves may be beneficial for minimising the exposure of the hoof and distal limb to large accelerations during hoof landing. At foot-off, it was most commonly observed that accelerations were amplified on the artificial surface compared to the turf; average accelerations per individual limb were 2–12% greater for the former. We inferred that the artificial surface deformed, at least to some extent, more elastically under load and subsequently recovered and returned a higher proportion of energy to the hoof. This will have aided propulsion, leading to more rapid hoof breakover, and it could confer a performance benefit. Overall, barefoot hooves typically experienced the lowest accelerations at foot-off; however, at top gallop speeds, accelerations for barefoot hooves appeared to increase at a relatively higher rate than for shod conditions. Further work is needed to relate these findings to injury risk and racing outcomes specifically, particularly in racehorses galloping at top speeds. 

## Figures and Tables

**Figure 1 animals-12-02161-f001:**
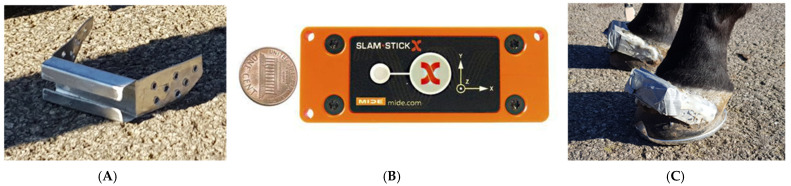
(**A**) Aluminium bracket. (**B**) Slamstick accelerometer. (**C**) Accelerometer and bracket system mounted to hoof.

**Figure 2 animals-12-02161-f002:**
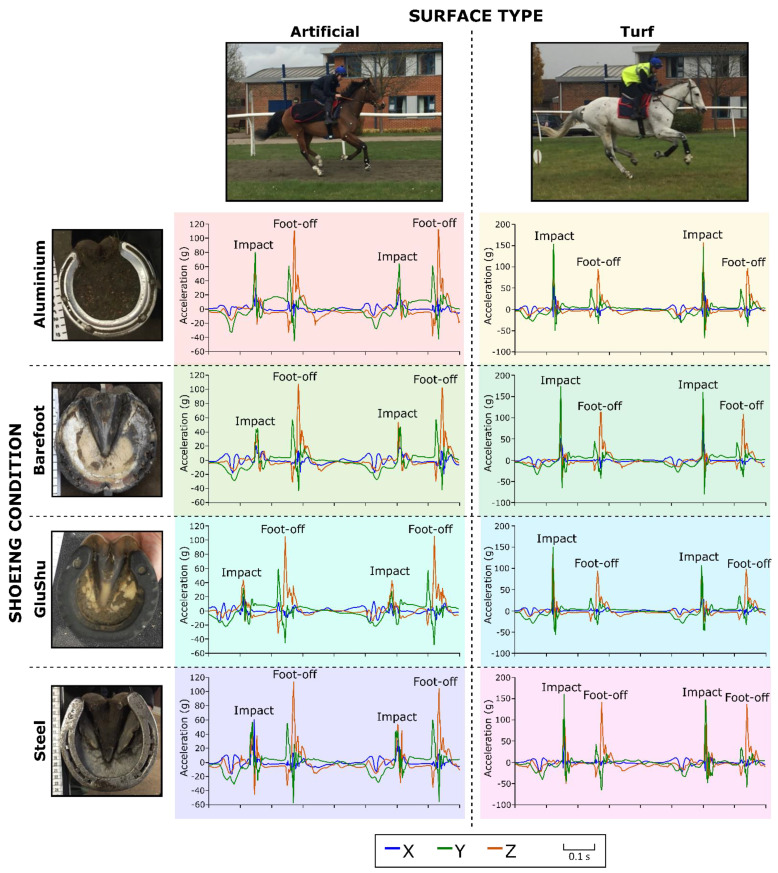
Examples of tri-axial accelerometry data collected from the hooves of the horse from horse–jockey combination 8 under the eight different shoe–surface combinations: x is medio-lateral, y is along the hoof wall proximo-distal and z is dorso-palmar.

**Figure 4 animals-12-02161-f004:**
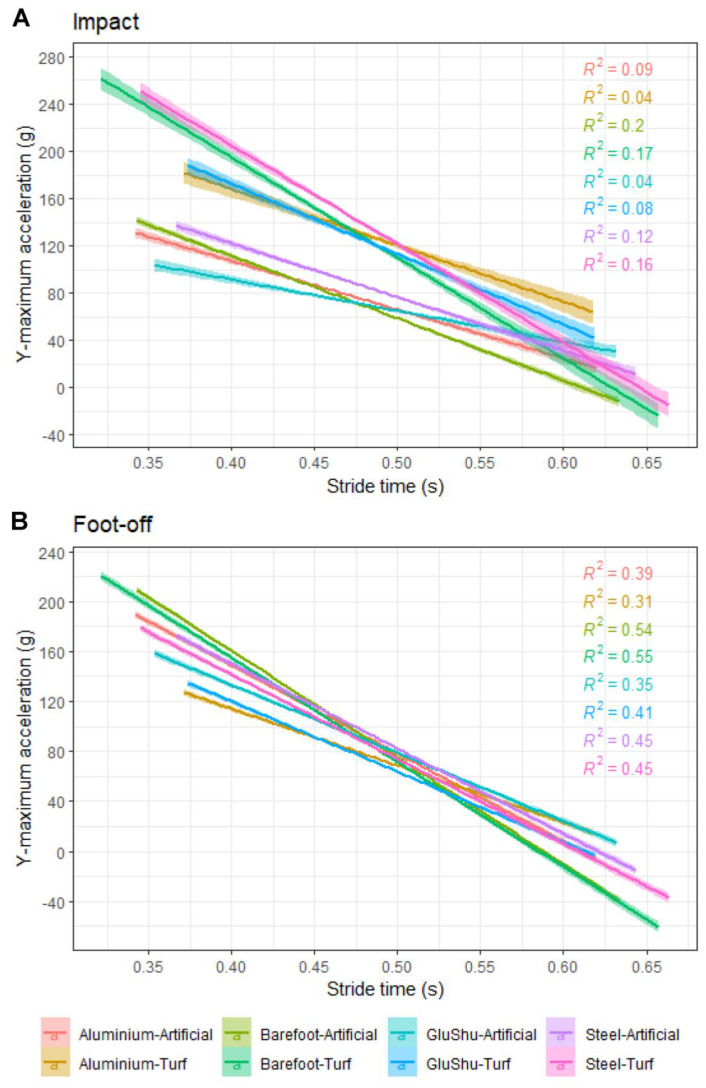
Relationship between hoof accelerations and stride time for the y-maximum parameter at (**A**) impact and (**B**) foot-off, subdivided by the shoe–surface combination for the entire dataset. The r^2^ values for the linear regressions are indicated. All *p* < 0.001.

**Table 1 animals-12-02161-t001:** Number of strides analysed per shoe–surface combination for each horse–jockey pair. Please note that the horse–jockey ID numbers are consistent with References [26,70,71].

Horse–Jockey Pair ID	Shoe–Surface Combination							
	Aluminium–Artificial	Aluminium–Turf	Barefoot–Artificial	Barefoot–Turf	GluShu–Artificial	GluShu–Turf	Steel– Artificial	Steel–Turf
1	403	376	435	365	354	360	477	345
2	389	574	826	341				
3	903	379	465	582	332	307	670	811
4	397	411	332	378	386	552	291	537
5	776		655					
6	506		344		371	590	423	
7	210		627		409		249	
8	665	479	1054	339	315	664	684	449
9			443		533			
10	499	605	480	488	637	534	425	538
11	922	432	473	299	616	504	915	361
13	571	622	414	350	416	495	404	544
14	382	350	271	275	389	331	294	365
15			319	586			384	930

**Table 2 animals-12-02161-t002:** Summary of peak accelerations at impact and foot-off across the x, y and z acceleration axes for each of the limb types. Please note that, alongside the leading and non-leading limb data, “mixed lead” runs were included in the forelimb and hindlimb data outputs listed as “combined”.

Limb	Acceleration Parameter	Impact Mean (g)	2 SD	Foot-Off Mean (g)	2 SD	*n*	Mean Stride Time	2 SD
All limbs	X-maximum	44.16	67.85	16.55	20.65	41,183	2399.15	344.46
	X-minimum	−21.79	35.13	−17.17	17.85	41,183		
	Y-maximum	98.31	127.82	89.22	72.21	40,844		
	Y-minimum	−34.27	54.32	−38.11	44.08	40,844		
	Z-maximum	88.73	128.02	49.93	41.13	40,691		
	Z-minimum	−38.83	49.25	−30.46	39.28	40,691		
Combined Forelimb	X-maximum	36.96	47.27	14.57	18.60	20,882	2403.59	350.90
	X-minimum	−19.62	28.35	−17.26	18.47	20,882		
	Y-maximum	87.68	99.21	95.46	66.70	20,543		
	Y-minimum	−33.89	51.21	−34.88	36.67	20,543		
	Z-maximum	89.23	120.27	45.76	40.39	20,882		
	Z-minimum	−37.13	45.88	−28.97	42.96	20,882		
Leading Forelimb	X-maximum	42.38	49.43	14.61	20.08	8781	2411.34	334.76
	X-minimum	−20.72	31.83	−16.69	16.91	8781		
	Y-maximum	94.42	97.16	95.70	64.88	8633		
	Y-minimum	−36.57	51.39	−34.53	34.30	8633		
	Z-maximum	102.19	123.69	46.83	44.98	8781		
	Z-minimum	−35.39	38.42	−29.86	43.14	8781		
Non-Leading Forelimb	X-maximum	30.72	37.07	13.32	14.47	8641	2405.54	342.77
	X-minimum	−17.37	22.84	−17.33	19.64	8641		
	Y-maximum	78.00	83.84	91.63	63.04	8450		
	Y-minimum	−27.69	38.68	−32.79	31.60	8450		
	Z-maximum	67.93	85.40	43.83	36.58	8641		
	Z-minimum	−36.64	48.57	−27.94	41.75	8641		
Combined Hindlimb	X-maximum	51.56	81.29	18.58	21.84	20,301	2394.59	337.47
	X-minimum	−24.02	40.47	−17.09	17.19	20,301		
	Y-maximum	109.06	148.30	82.91	75.31	20,301		
	Y-minimum	−34.65	57.29	−41.38	49.64	20,301		
	Z-maximum	88.20	135.71	54.32	40.06	19,809		
	Z-minimum	−40.62	52.34	−32.03	34.71	19,809		
Leading Hindlimb	X-maximum	59.38	85.87	19.82	22.62	8719	2405.71	321.97
	X-minimum	−27.07	42.25	−16.27	17.18	8719		
	Y-maximum	123.44	150.61	77.82	72.50	8719		
	Y-minimum	−37.45	55.64	−39.76	46.93	8719		
	Z-maximum	98.53	120.37	51.75	36.95	8606		
	Z-minimum	−41.37	62.88	−30.21	35.67	8606		
Non-Leading Hindlimb	X-maximum	39.53	57.21	16.25	19.08	8384	2401.12	321.97
	X-minimum	−19.05	32.45	−17.20	14.84	8384		
	Y-maximum	87.07	120.04	83.80	67.21	8384		
	Y-minimum	−30.82	52.33	−40.37	45.89	8384		
	Z-maximum	65.58	89.33	55.99	40.33	8223		
	Z-minimum	−39.20	39.22	−33.47	32.39	8223		

## Data Availability

Data supporting the results are presented in the Results section of this manuscript and the Appendix A.

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
