# Peer review of "Hoof Impact and Foot-Off Accelerations in Galloping Thoroughbred Racehorses Trialling Eight Shoe–Surface Combinations"

_animals, 2022, doi:10.3390/ani12172161_

Round 1
Reviewer 1 Report
This manuscript “Hoof impact and foot-off accelerations in galloping Thorough-bred racehorses trialling eight shoe-surface combinations” represent research with high clinical relevance. The reviewer congratulates the authors for such a well-designed study and such a valuable outcome. This research gives a very important input to the racing industry and the examinations have been performed with high effort. Unfortunately, the manuscript is very much too long. All sections are well structured and the content is interesting and fits to the performed research. However, the manuscript has less the character of a scientific article, but more as a thesis. I do not know how the editorial board deals with such a long manuscript and leave it up to the editors to make a final decision. Still, I recommend shortening of the manuscript and focus on clear hypotheses and main findings.
Simple summary
It is well written and includes all relevant information.
Abstract
It is well written and includes all relevant information.
Introduction
The introduction is very long. It is well structured and includes a lot very interesting aspects and information, but usually the introduction is approximately 500 words long. You do not have to shorten it down to this word count, but I would recommend to focus a bit and delete some less necessary information. For example, the section from line 81-92 is not of essential relevance (“However, excessive hoof slide…and overtake the position of the upper body while travelling at high speed.”). Please try to shorten and focus as much as possible in each section with regard to clinical relevance, effect of race track surface and shoeing. You can also stop the introduction after the conclusion of the aim of the study. It is a shame but delete line 189-210.
In addition I miss the statement of clear hypotheses, which could also be used as a tool to focus better on the relevant outcome and shorten the manuscript.
Material and methods
There was no orthopedic check of the horses by veterinarians before the measurements?
Please just state, if there were severe differences in climate/weather on the different examination days in the text.
It is hard to imagine the sensors at the dorsal wall of the hind hooves were not destroyed or lost due to collision with the front hooves.
Was the running speed measured and considered during the analysis? In my understanding differences in speed between the different compared conditions also influence the acceleration. How has the speed been standardized or at least speed variations considered in the analysis?
How many strides per horse have been measured and included in the averaged number of strides? And why do you have so severe variations in the number of analyzed strides between the different conditions? (1484 – 599)?
Results
I am sorry, but it is much too long. It is well structured and the findings are well represented and the structure of this section is absolutely clear, but I am not sure if the journal will publish such a high word count. It would advisable to either dramatically shorten to tables and data/findings to the absolute minimum or it is questionable if you would benefit from dividing this paper in two or three papers. For example: 1st paper: general pattern in acceleration independent of shoeing and surface, 2nd paper: shoe effect, 3rd paper: track effect
Discussion
This section is also more than double that long as usual. It is easy to follow and again well structured, but too long. It would be better to focus on the main findings.
References
The number of references is also double the number as usual.
Author Response
Please refer to the attached file -thank you.

Reviewer 2 Report
This paper describes the impact and lift off accelerations of galloping horses with 4 different types of shoeing and on two different tracks.
Generally the paper is of high technical standard, the study design and analysis are sound and the aim of the study interesting.
However, despite the lack of a word limit by animals (“Animals has no restrictions on the length of manuscripts, provided that the text is concise and comprehensive.”), the text is much too long and not concise. Introduction, results and discussion are far too long and do not lead to the relevant points. There are also extreme detailed tables, which might be useful as data base in a supplement, but not for an journal paper.
Therefore I would strongly recommend to shorten the paper to at least one third and condense the results for the interested reader.
Some minor questions or comments:
Please us only SI units (line 225)
Please describe the method of defining the positions of impact and foot off more in detail and with corresponding images.
Author Response

(The authors gave the same response as above.)

Reviewer 3 Report
The paper covers a very interesting and highly practical subject. The overall view of the paper seems sound. However, two main major problems should be corrected.
1. the analysis – there are many single comparisons done - there are so many that they are not readable. It is not possible even in an electronic journal to publish tables for three pages or to publish figures with results that are not even 2cmx2cm. These results are not visible at all. They are not visible as illustrations on the page and they are also not visible as the whole information. Why you do not use analysis of variance with all factors? Using analysis of variance you will receive clear factor influence corrected for all other effects. You will be able to receive direct information on differences between for example aluminum and steel with the corrections for all other effects – also surface ground.
2. some parts are too long:
– introduction - is very interesting, however many details – only specialists in the field will manage to follow,
- results are doubled – all are included in long tables and described, figures so small that they are not informative,
The paper needs re-writing. If the analysis of variance will be used all information will be much easier to present as tables for each factor – shoe, surface, stance phase, limb, or axis.
All other information can be used as supplementary, however, it should be also combined to be clear enough.
Author Response
Please refer to the attached file - thank you.

Reviewer 4 Report
This manuscript is very well written. I was able to follow the entire research question. I had to review the article again to find any technical errors but the methodology is sound and well-oriented.
However, there are some text mistakes in the introduction part with some very long sentences and mistakes in the referencing style. I would suggest reviewing your introduction once again and try to decrease the length of the introductory part if possible. Also, please consider transferring your tables to supplementary data. In my opinion, graphs are enough for the reader to follow the trend.
Author Response

(The authors gave the same response as above.)

Round 2
Reviewer 1 Report
Dear authors,
Thanks for shortening and re-structuring the manuscript! As far as I see you considered most of my comments. Since I see the great effort you made and since you have my greatest respect for the study design and measurements I accept the manuscript in its current form!
Kind regards
Author Response
Thank you for your positive comments and for taking the time to review our manuscript.